human–computer interaction

dreams, NLP, Hall–Van de Castle, digital health, social computing

**Author for correspondence:**
Luca Maria Aiello
e-mail: lajello@gmail.com

# Our dreams, our selves: automatic analysis of dream reports

Alessandro Fogli[1], Luca Maria Aiello[2]
and Daniele Quercia[2]

[1]Computer Science Department, Università degli studi di Roma Tre, Rome, Italy
[2]Nokia Bell Laboratories, Cambridge, UK

LM, 0000-0002-0654-2527

Sleep scientists have shown that dreaming helps people improve their waking lives, and they have done so by developing sophisticated content analysis scales. Dream analysis entails time-consuming manual annotation of text. That is why dream reports have been recently mined with algorithms, and these algorithms focused on identifying emotions. In so doing, researchers have not tackled two main technical challenges though: (i) how to mine aspects of dream reports that research has found important, such as characters and interactions; and (ii) how to do so in a principled way grounded in the literature. To tackle these challenges, we designed a tool that automatically scores dream reports by operationalizing the widely used dream analysis scale by Hall and Van de Castle. We validated the tool's effectiveness on hand-annotated dream reports (the average error is 0.24), scored 24 000 reports—far more than any previous study—and tested what sleep scientists call the 'continuity hypothesis' at this unprecedented scale: we found supporting evidence that dreams are a continuation of what happens in everyday life. Our results suggest that it is possible to quantify important aspects of dreams, making it possible to build technologies that bridge the current gap between real life and dreaming.

## 1. Introduction

Research has repeatedly provided strong support for what sleep scientists refer to as the 'continuity hypothesis of dreams': most dreams are a continuation of what is happening in everyday life. It turns out that everyday life impacts dreaming (e.g. anxiety in life leads to dreams with negative affect) [1,2], and vice versa (e.g. dreaming impacts problem-solving skills) [3,4].

In the therapeutic context, the main goal of dream analysis is to help people address their real-life problems. This hypothesis provides a theoretical basis for therapy as it can

be used to raise self-awareness, to identify latent emotional states, and to help people cope with significant life events and traumas. Given that, dream analysis is used to address many mental health issues. For those suffering from nightmares, interpreting dreams and ultimately influencing them are ways of partly treat their condition. To see how, consider the concept of lucid dreaming. A lucid dream is one in which the dreamer is aware that they are dreaming, and can control their actions [5]. For such dreams, being able to interpret them translates into being able to influence them, and such an ability is often used as a treatment: therapies based on lucid dreaming have repeatedly been found to be effective in reducing nightmare frequency [6–8]. In contemporary therapy, imagery rehearsal therapy (IRT) is often used [9]: therapists ask their clients to recall their bad dreams, write them down and then change their content to something positive, encouraging their clients to mentally rehearse these new dream scripts every day to decrease the frequency and intensity of the nightmares.

More generally, dream interpretation is one tool that has been used by therapists for a long time [5,10–14]. For example, it has been used for exploring wish fulfillment, unconscious desires and conflicts (in Freudian and Jungian analyses), and for bringing dream content into a client's actual life, helping the client clarify feelings from all angles (in Gestalt therapy).

Because dream interpretation helps people improve their waking lives, sleep scientists have developed increasingly sophisticated ways of coding dreams. Winget & Kramer reviewed 150 dream rating and content analysis scales [15], and found that the best validated and most widely used scale remains Hall & Van de Castle's [16,17].

As we shall see in the Background section (§2), dream content analysis scales are complex and, as such, require human intervention. As a result, annotations have been mostly done manually, which is time-consuming, does not scale, and cannot be often used in technologies (e.g. in a mobile phone app keeping track of dreams and analysing them on-the-fly). So far, most attempts of automating dream analysis have mainly focused on identifying emotions [18–21] and have not captured other aspects that dream research has found important, such as characters and their interactions.

To partly fill the gap, we designed a way to test the 'continuity hypothesis' at scale. In so doing, we made four main contributions:

(i) we set out to test the 'continuity hypothesis' by formulating five hypotheses that specifically associated waking life with dreaming (§3);

(ii) we designed and validated a tool that is able to automatically score dream reports (§4). After operationalizing the most important factors of Hall and Van de Castle's dream coding scale (§4.1) and collating our dream reports (§4.2), we built a tool that was able to process such reports (§4.3). We validated the tool's effectiveness on a previously annotated set of dream reports (§4.4): the average error was 0.24; and

(iii) we then scored a number of dream reports (24 000) far larger than any of those in previous studies and, given this unprecedented scale, we were able to test our five hypotheses (§5). We found supporting evidence for the 'continuity hypothesis': differences in dream reports' markers systematically mirrored differences typically found in the real world, and that held for a variety of factors, including gender, age, life-changing experiences and even the impact of societal aggression.

# 2. Background

## 2.1. Origins of dream interpretation

The notion that dreams contain hidden meanings has been popular for centuries. During the second century AD, Artemidorus Daldianus produced a five-volume work entitled *Oneirocritica* (The Interpretation of Dreams) [22,23], which catalogued the meaning of a large number of symbols and situations. Artemidorus' volumes were the most popular work on dream interpretation until modern times. Only another interpretation system—a recent one—enjoyed a similar success. This 1890s system was one of Sigmund Freud's best-known works, '*The interpretation of dreams*' [24]. It attempted to associate specific meanings to characters, objects, animals and scenarios that frequently appeared in dreams. Despite recognizing the importance of a large number of symbols and situations, Freud mainly focused on two basic hidden human needs: sex and aggression.

## 2.2. Modern dream coding

Sleep scientists have recently shown that dreams contain hidden messages but these messages are not to be reduced to sex and violence [25]. They have done so by developing increasingly sophisticated ways of coding dreams. There are more than 150 dream rating and content analysis scales [15], and the best validated and most widely used one remains Hall & Van de Castle's [16,17]. To put it simply, this coding system sees a dream as: (i) a cast of *characters* (e.g. depressed man, friendly woman, a cute dog); (ii) a plot in which characters *interact* with each other (e.g. the depressed man patting the dog); and (iii) a process reflecting different affective states (e.g. the depressed man now feels safe and relaxed). Using this content analysis scale, William Domhoff collected and analysed thousands of dream reports [26]. Domhoff's work and countless of subsequent studies have provided strong support for what sleep scientists refer to as the 'continuity hypothesis' [27]: most dreams are a continuation of what is happening in everyday life. This hypothesis is consistent with the 'simulation' theories of dreaming, according to which threats perceived in everyday life (increased anxiety) lead to increased threat simulation in dreams (increased negative affect). A recent social simulation theory even suggested that dreaming helps simulate social skills and, in so doing, strengthens real-life social relationships [28]. Overall, researchers were able to extensively test the 'continuity hypothesis' only for a specific class of individuals: clinical patients. Researchers found that people with different mental health disorders, such as anxiety and depression, sleep disorders and health behaviour problems reported more nightmares [1,2]. The collection of dream reports has been focused on clinical patients simply because such a collection is easier.

## 2.3. Dreams as therapy

There is, however, one notable difference between dreams and everyday life. Regardless of a person's everyday experiences, 80% of dreams involve some forms of negative emotions. This considerable presence of negative emotions had been explained with what sleep scientists called the 'nocturnal therapist' theory [29,30]. According to this theory, dreams can be seen as a 'nocturnal therapist' that helps a dreamer identify worries and concerns. The 'nocturnal therapist' has also been found to help find innovative solutions to problems: while dreaming, the mind attempts to solve a problem by looking at it from unusual perspectives [3,4]. The only quantitative work on the nocturnal therapist hypothesis focused on dream affect [31], and showed that individuals whose dream reports contained more positive affect and emotions also exhibited higher well-being and life satisfaction.

## 2.4. Our contribution

To sum previous work up, the results in the literature have to be treated with caution owing to relatively small sample sizes, an over-simplified approach to dream interpretation focused on affect, and an over-focus on clinical patients. To partly fix this gap, we collated more than 38 000 dream reports, built a tool to automatically interpret these reports in terms of not only affect but also characters and interactions in line with Hall and Van de Castle's scale, and, for the first time, tested the 'continuity hypothesis' at scale.

# 3. Research hypotheses

In support of the continuity hypothesis, we can observe that about 80% of dreams are experienced from a first-person perspective (from the dreamer's perspective) and tend to involve ordinary scenarios [26].

In turn, everyday life is systematically affected by a variety of factors, including one's sex, phase of life, (life-changing) experiences and exposure to everyday aggression. Our dataset allows for testing the continuity hypothesis for all those factors. More specifically, we can study five factors: (i) sex differences as our data reports a dreamer's sex; (ii) adolescence as a phase of life as our data includes a young adult named Izzy who recorded her dreams between the ages of 12 and 25; (iii,iv) war and blindness as two life-changing personal experiences as our data contains dream reports of a war veteran and of a few people who became blind; and (v) everyday aggression as our data contains dream reports from the 1960s to 2000s, and these decades were characterized by different levels of violence.

We selected these factors because they are present in theoretical work [26,27,31], and our data make it possible to test them. Having these five factors, we tested whether the ways dreams are mediated by them are similar to the ways the literature has reported everyday life to be mediated by them. If they are similar, then there would be supporting evidence for the continuity hypothesis. Next, we

summarize the literature that relates everyday life to the characteristics quantifiable from dream reports (e.g. positive and negative emotions, aggressiveness), and do so one factor at a time.

## 3.1. Sex

Cultural and psychological studies have repeatedly shown that aggressive behaviour is less frequent and less intense in women than that in men [32]. Those differences might result from evolutionary factors [33], and from social, cultural, and economic factors [34]. Also, the ability to express emotions, which has been shown to relieve stress and increase well-being [35], is yet another factor impacting aggressiveness. That ability is more developed in women than that in men [36,37]. It has been shown that female friends typically talk about their emotions while male friends are more accustomed to suppressing theirs [38]. As a result, men do not get emotional support from each other but rather 'do stuff' together (e.g. golf, ski, drink) [38]. The combined product of genetics and cultural factors has been generally associated with sex differences in aggressive behaviour and emotion expression, and these associations have held in a number of scenarios [39–41]. In 2007, the manual inspection of a small-scale corpora of dream reports indeed showed that, as opposed to women, 'men dream about physical aggression more often' [42]. For the continuity hypothesis, it follows that:

**H1** female's dreams are characterized by emotions rather than interactions around activities, and by limited levels of aggression.

## 3.2. Adolescence

As children reach adolescence, their emotional stability is put to test. Social anxiety increases owing to a variety of factors including exposure to new social experiences, increased conflicts with parents and hormonal changes [43–45]. Experiencing negative emotions by adolescents has been systematically recorded for both sexes, especially for females [46]. Girls between the ages of 10 and 14 typically show an initial dip in emotional stability, followed by an increase as they reach young adulthood and start to engage in sexual interactions [47]. Then our second hypothesis is:

**H2** an adolescent's dreams are characterized by negative emotions, followed by sexual interactions in early adult life.

## 3.3. War

About 17% of American troops returning from Iraq and 11% of those coming back from Afghanistan have suffered from post-traumatic stress disorder (i.e. an anxiety disorder caused by very stressful, frightening or distressing events) [48]. Our third hypothesis is then:

**H3** a war veteran's dreams are characterized by negative emotions and aggression. This hypothesis was found to be true among Vietnam veterans: they were found to frequently experience dreams associated with guilt and violence [49].

## 3.4. Blindness

Blind people's daily lives have been found to be comparable to those of individuals in the general population [50], yet there are two main exceptions. First, blind people might need carers, which often happen to be women [51–53], and, as such, the continuity hypothesis suggests that carers would be present in blind people's dream reports. Second, blind people's sensory perceptions tend to be heightened because of an increased sensitivity in tactile, acoustic and olfactory qualities [54]. This sensitivity has an effect on the inner workings of imagination, so much so that previous studies showed blind people envision more 'unreal' imagery than sighted people [55]. This matches the dependency thesis [56], which states that there is a conceptual connection between perception and sensory imagination. Based on these elements, our fourth hypothesis is then:

**H4** blind people's dreams feature more imaginary characters and aspects related to their real-life carers (e.g. they feature women).

## 3.5. Everyday aggression

The level of violent crime in USA was considerable in the 1960s and then steadily declined, and continues to decline today [57]. Our fifth and final hypothesis is then:

**H5** dreams during times of societal aggression (in the 1960s) are characterized by aggression. This final hypothesis was tested for the September 11th terrorist attacks on the Twin Towers in New York City. It has been shown that these attacks caused a dramatic increase in the number of people *across US* dreaming about explosions, death and fire [58]. So not only experiencing but also hearing about a stressful event can result in highly traumatic dreams.

In summary, we formulated five hypotheses:

**H1** female's dreams are characterized by emotions rather than interactions around activities, and by limited levels of aggression;

**H2** an adolescent's dreams are characterized by negative emotions, followed by sexual interactions in early adult life;

**H3** a war veteran's dreams are characterized by negative emotions and aggression;

**H4** blind people's dreams feature more imaginary characters and aspects related to their real-life carers (e.g. they feature women); and

**H5** dreams during times of societal aggression (in the 1960s) are characterized by aggression.

# 4. Methods

After introducing the theoretical framework for dream coding (§4.1), we provide an overview of our tool's data sources (§4.2) and the dream coding's operationalization (§4.3).

## 4.1. Dream coding system

In the 1940s, psychologist Calvin Hall analysed thousands of written dream reports, and he gradually developed empirical categories that he systematically measured in those reports. Later, with the help of fellow psychologist Robert Van de Castle, he expanded, refined, and formalized the categorization which was published in their book: '*The content analysis of dreams*' [16]. The final Hall–Van de Castle dream coding system consists of 10 categories (and their sub-categories) of elements appearing in dreams, together with detailed rules to recognize and measure those elements from written reports. In the following years, the system became a standard reference for quantitative dream analysis, also thanks to its objective approach that facilitates reproducibility and high inter-coder reliability. The 10 categories are:

— *characters:* people, animals and other figures featured in the dream;
— *interactions:* social interactions among characters (e.g. kissing);
— *emotions:* emotions experienced by the characters or denoting the situation (e.g. sadness);
— *activities:* physical actions that characters perform and sensory experiences (e.g. smelling);
— *striving:* success and failure of characters in carrying out their activities;
— *(mis)fortunes:* fortune and misfortunes that happen to the characters, possibly as result of their actions;
— *settings and objects:* physical surroundings or objects present in the scene (e.g. outdoors, a weapon);
— *descriptive elements:* attributes and qualities of objects, people and actions (e.g. colour, size, speed);
— *food and eating:* presence of food or act of eating; and
— *elements from the past:* characters or elements belonging to the dreamer's past (e.g. younger self);

In practice, these categories are not all of equal importance in capturing the psycho-pathological aspects of a dream's content. Dream scientists determined that the three categories of characters, social interactions and emotions are the most valuable ones [59] and are usually more informative than all the remaining ones combined. In this work, to test our five hypotheses, we focused on coding each dream report along measures that reflect these three categories (table 1).

(i) *Characters.* People, animals and imaginary figures who appear in the dream report. In addition to determining characters' types, we classified their gender and whether they were dead.
*Male%.* The number of male characters divided by the sum of male and female characters.
*Animals%.* The number of animal characters divided by the number of all characters.
*Imaginary%.* The number of fictional or dead characters divided by the number of all characters.

**Table 1.** The Hall–Van de Castle categories computed by our tool from dream reports. (The categories are listed with corresponding formulae (detailed in §4.3) and external resources used by the tool to compute them.)

|  | indicators | formula | implementation |
|---|---|---|---|
| characters (C) | Male% | $\dfrac{\lvert C_{Male}\rvert}{\lvert C_{Male}\rvert + \lvert C_{Female}\rvert}$ | Wikidata, Wordnet |
|  | Animals% | $\dfrac{\lvert C_{Animals}\rvert}{\lvert C\rvert}$ | Wikidata, Wordnet |
|  | Imaginary% | $\dfrac{\lvert C_{Dead}\rvert + \lvert C_{Fictional}\rvert}{\lvert C\rvert}$ | Thesaurus, Wikidata |
| interactions (I) | $F/C$ Index (friendly) | $\dfrac{\lvert I_{Friendly}\rvert}{\lvert C\rvert}$ | Wordnet |
|  | $S/C$ Index (sexual) | $\dfrac{\lvert I_{Sexual}\rvert}{\lvert C\rvert}$ | Wordnet |
|  | $A/C$ Index (aggressive) | $\dfrac{\lvert I_{Aggression}\rvert}{\lvert C\rvert}$ | Wordnet |
|  | Aggression% | $\dfrac{\lvert I_{Aggression}\rvert}{\lvert I\rvert}$ | Wordnet |
|  | Aggression/Friendliness% | $\dfrac{\lvert I_{Aggression}\rvert}{\lvert I_{Friendly}\rvert}$ | Wordnet |
| emotion words (W) | Negemo% | $\dfrac{\lvert W_{negative}\rvert}{\lvert W_{positive}\rvert + \lvert W_{negative}\rvert}$ | Emolex |

(ii) *Interactions.* Interactions among characters of three types: friendly, sexual and aggressive.
   *F/C index.* The number of friendly interactions divided by the total number of characters.
   *S/C index.* The number of sexual interactions divided by the total number of characters.
   *A/C index.* The number of aggressive interactions divided by the total number of characters.
   *Aggression.* The number of aggressive interactions divided by the total number of interactions.
   *Aggression/friendliness.* The number of aggressive interactions divided by the total number of friendly interactions.

(iii) *Emotions.* Markers of positive or negative emotions in the dream report.
   *Negative emotions%.* The ratio between the number of negative emotions and the total number of emotions expressed in the dream.

The Hall–Van de Castle coding system categorizes and counts the elements appearing in dreams but does not provide any composite metric to directly inform a psycho-pathological interpretation of the dream's content. In his book '*Finding meaning in dreams*' [26], psychologist William Domhoff addressed this limitation by combining the counts from the dream coding systems into simple measures which have proven to correlate with what dreamers were experiencing in their daily lives [60–63]. To spot anomalies in the content of a dream, one needs to compare a given dream report's values for the metrics in table 1 against a 'typical' dream report's values. In the literature on dreams, Cohen's $h$ has become the standard way of doing that [26]. In statistics, Cohen's $h$ is a measure of distance between two proportions [64]. Given the measured proportion $p$ and the norm proportion $p_{norm}$, $h$ is calculated as:

$$h = (2 \cdot \arcsin(\sqrt{p})) - (2 \cdot \arcsin(\sqrt{p_{norm}})). \tag{4.1}$$

When $h$ for a measured proportion is positive, then the proportion is higher than the norm; when negative, it is lower. This measure comes with a rule-of-thumb interpretation of its ranges, which segments the distance to be small ($h > 0.2$), medium ($h > 0.5$), or large ($h > 0.8$). In clinical settings, it has been found that the more one's dreams deviate from the norms, the more likely the dreamer is affected by latent or manifest psycho-pathologies [26]. The $h$-value measures the difference between the proportion computed on a given dream set and the same proportion computed on the normative set. To also measure the statistical significance of this difference, we followed the standard practice in previous work [26] and ran a paired $t$-test between each proportion computed on a given dream set and the same proportion computed on the normative set. The resulting $p$-values allowed us to estimate whether the mean difference between the proportions in the two sets was close to zero.

**Table 2.** Summary of the dream reports under study. (For some reports, the dreamers' identities are unknown; therefore, for some dream sets, the lower bound for the number of unique dreamers is reported (denoted with a '+' suffix added to the number of dream reports, that is, [no. *dreamers*] +).)

| dream set | dreamer characteristics | sex | decade | no. dreamers | no. dreams |
|---|---|---|---|---|---|
| full set | all dreamers | m ± f | 1910–2010 | 500 ± | 24 035 |
| no-condition set | no-condition dreamers | m ± f | 1910–2010 | 480 ± | 23 042 |
| male set | male dreamers | m | 1910–2010 | 180 ± | 6790 |
| female set | female dreamers | f | 1910–2010 | 300 ± | 16 252 |
| blindness set | blind dreamers | f | 1990 | 5 | 238 |
| | blind dreamers | m | 1990 | 10 | 143 |
| | Edna (blind woman) | f | 1940 | 1 | 19 |
| war veteran set | Vietnam veteran | m | 1970–2010 | 1 | 593 |
| annotated set | volunteer dreamers | m | 1960–1990 | 25 | 1700 |
| | volunteer dreamers | f | 1960–1990 | 21 | |
| | volunteer dreamers | — | 1960–1990 | 4 | |
| normative set | volunteer dreamers | m | 1940–1950 | 100 | 490 |
| | volunteer dreamers | f | 1940–1950 | 100 | 491 |
| Izzy set | Izzy age 12 | f | 2000 | 1 | 52 |
| | Izzy age 13 | | 2000 | | 167 |
| | Izzy age 14 | | 2000 | | 249 |
| | Izzy age 15 | | 2000 | | 539 |
| | Izzy age 16 | | 2000 | | 845 |
| | Izzy age 17 | | 2000 | | 741 |
| | Izzy age 18–21 | | 2010 | | 1359 |
| | Izzy age 22–25 | | 2010 | | 377 |

## 4.2. Resources and data

### 4.2.1. Dreambank

DreamBank.net [65] is the largest online public repository of written dream reports. It contains over 38 000 dream descriptions gathered from a variety of verified sources and research studies. Dream reports are annotated with their dates of recording, which span six decades (from 1960 to 2015), and are linked to free-text descriptions of the dreamers, which contain information about their gender, age (ranging from 7 to 74), profession and personal history. We collected all the DreamBank reports and created seven sets of dreams (whose summary statistics are reported in table 2).

*Full set.* A set of 24 k dream reports obtained by selecting all the reports written in English and composed by at least 50 words from the initial corpus of 38 000 reports. The choice of 50 words ensures a minimal length for text mining as indicated by the original Hall & Van de Castle guidelines [16]. Given the uneven distribution of reports across decades (figure 1), for our temporal analyses, we could not go at a level finer than 'decade' and, as such, we computed confidence intervals to ascertain the reliability of the results for the less-represented decades.

*Blindness set.* A set of 400 dreams reported by people affected by complete vision loss either from birth or for over 20 years.

*War veteran set.* A set of 600 dream reports recorded from 1971 to 2017 by a Vietnam war veteran who had a very intense and traumatic experience of that conflict.

*No-condition set.* A set of 23 000 reports obtained by filtering away from the *full set* all the reports in the *blindness set* and the *war veteran* set. In this 'no-condition set', we included only the individuals for which no apparent pathology was reported based on the short dreamer descriptions attached to most dream reports. Given its considerable size, this set might allow for comparing the dreams of individuals with special conditions with those of individuals in the general population.

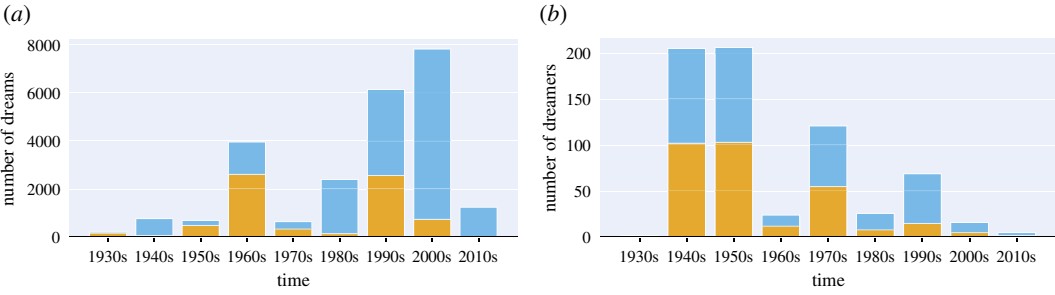

**Figure 1.** (*a*) Total number of dream reports and (*b*) unique dreamers in the *full set* for each decade. Male dreams and dreamers are displayed in orange, female in blue.

*Izzy set.* A set of more than 4300 dream reports recorded by Izzy, a young woman passionate about collecting her dreams, over the course of 13 years from age 12 to age 25.

*Normative set.* A set of 1000 dream reports hand-coded by Hall and Van de Castle themselves in the late 1940s and early 1950s [16]. Two hundred American university students (100 male and 100 female) in Cleveland (Ohio) were asked to write down five dreams each. Normative values in dream studies are generally based on this set.

*Annotated set.* A set of 1700 dream reports from 50 people (21 female, 25 male and four unknown) collected over four decades at different university campuses [26]. A team of experts coded all the dream reports according to the Hall–Van de Castle system.

### 4.2.2. Knowledge bases

Our feature extraction relied on two external knowledge bases: Wikidata and WordNet. Wikidata [66] is a collaboratively edited knowledge base maintained by the Wikimedia Foundation. It contains 55M ± unique data items—which represent topics, concepts, or objects—annotated with a main textual label, a description and other supporting metadata. These items are interlinked by means of labelled semantic relationships called *properties*. For example, the item labelled as *Elvis Presley* is connected to the node *Male* by the property *instance of* .

WordNet [67] is a lexical database for the English language. Nouns, verbs, adjectives and adverbs are grouped into sets of synonyms called *synsets*, each expressing a distinct concept. Synsets come with metadata, including their categories which are called 'lexical domains' (a domain is a category such as *noun.person* and *verb.motion* among the possible 45 categories arranged in a two-level taxonomy).

## 4.3. The dream processing tool

Having the dream reports and the two knowledge bases at hand, we built our dream processing tool (figure 2). Next, we describe how the tool pre-processes each dream report (§4.3.1), and then identifies characters (§4.3.2, §4.3.3), social interactions (§4.3.4) and emotion words (§4.3.5). We chose to focus on these three dimensions out of all the ones included in the Hall–Van de Castle coding system for two reasons. First and foremost, these three dimensions are considered to be the most important ones in aiding the interpretation of dreams, as they define the backbone of a dream plot [26]: who was present, which actions were performed and which emotions were expressed. These are, in fact, the three dimensions that traditional small-scale studies on dream reports mostly focused on [68–70]. Second, some of the remaining dimensions (e.g. success and failure, fortune and misfortune) represent highly contextual and potentially ambiguous concepts that are currently hard to identify with state-of-the-art natural language processing (NLP) techniques, so we will recommend research on more advanced NLP tools as part of future work.

### 4.3.1. Preprocessing

The tool initially expands all the most common English contractions[1] (e.g. 'I'm' to 'I am') that are present in the original dream report. That is done to ease the identification of nouns and verbs. The tool does not remove any stop-word or punctuation to not affect the following step of syntactical parsing.

---

[1]List of English contractions: https://en.wikipedia.org/wiki/Wikipedia:List.of.English.contractions.

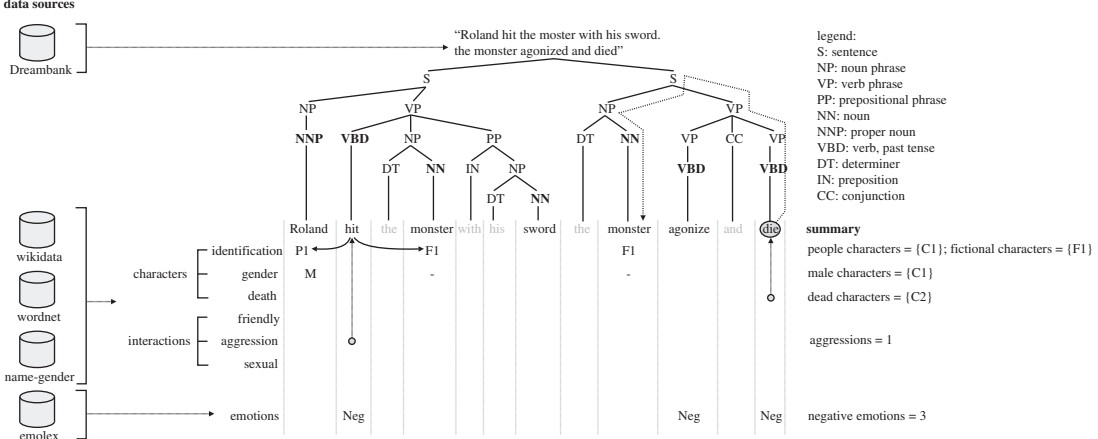

**Figure 2.** Application of our tool to an example dream report. The dream report comes from Dreambank (§4.2.1). The tool parses it by building a tree of verbs (VBD) and nouns (NN, NNP) (§4.3.1). Using the two external knowledge bases, the tool identifies people, animal and fictional characters among the nouns (§4.3.2); classifies characters in terms of their sex, whether they are dead, and whether they are imaginary (§4.3.3); identifies verbs that express friendly, aggressive and sexual interactions (§4.3.4); determines whether each verb reflects an interaction or not based on whether the two actors for that verb (the noun preceding the verb and that following it) are identifiable; and identifies positive and negative emotion words using Emolex (§4.3.5).

On the resulting text, the tool applies *constituent-based analysis* [71], a technique used to break down natural language text into its constituent parts that can then be later analysed independently. Constituents are groups of words behaving as coherent units which belong either to phrasal categories (e.g. noun phrases, verb phrases) or to lexical categories (e.g. nouns, verbs, adjectives, conjunctions, adverbs). Constituents are iteratively split into subconstituents, down to the level of individual words. The result of this procedure is a *parse tree*, namely a dendrogram whose root is the initial sentence, edges are production rules that reflect the structure of the English grammar (e.g. a full sentence is split according to the subject–predicate division), nodes are constituents and sub-constituents, and leaves are individual words.

Among all the publicly available techniques for constituent-based analysis, our tool incorporates the StanfordParser [72] from the nltk python toolkit [73], a widely used state-of-the-art parser based on probabilistic context-free grammars [71]. The tool outputs the parse tree and annotates nodes and leaves with their corresponding lexical or phrasal category (top of figure 2).

After building the tree, by then applying the morphological function *morphy* in nltk, the tool turns all the words contained in the tree's leaves into corresponding lemmas (e.g. it turns 'dreaming' into 'dream'). To ease understanding of the next processing steps, table 3 reports a few processed dream reports.

### 4.3.2. Identification of characters

To identify the characters mentioned in the dream report, we first built a database of nouns referring to the three types of actors considered by the Hall–Van de Castle system: people, animals and fictional characters.

To gather a list of people names, we merged the set of Wordnet words under the lexical domain of *noun.person* with the words that are *subclass of* or *instance of* the item *Person* in Wikidata. Similarly, for animal names, we merged all the words under the *noun.animal* lexical domain of Wordnet with the words that are *subclass of* or *instance of* the item *Animal* in Wikidata. To identify fictional characters, we considered the words that are *subclass of* or *instance of* the Wikidata items *Fictional Human*, *Mythical Creature* and *Fictional Creature*. As a result, we obtained three disjoint sets containing nouns describing people $N_{\text{People}}$ (25 850 words), animals $N_{\text{Animals}}$ (1521 words) and fictional characters $N_{\text{Fictional}}$ (515 words). These three sets contain both common nouns (e.g. fox, waiter) and proper nouns (e.g. Jack, Gandalf). Dead and fictional characters are grouped into a set of *Imaginary* characters ($C_{\text{Imaginary}}$).

Having those three sets, the tool is able to extract characters from the dream report. It does so by intersecting these three sets with the set of all the proper and common nouns contained in the report ($N_{\text{Dream}}$). In so doing, the tool extracts the full set of characters $C = C_{\text{People}} \cup C_{\text{Animals}} \cup C_{\text{Fictional}}$, where $C_{\text{People}} = N_{\text{Dream}} \cap N_{\text{People}}$ is the the set of person characters, $C_{\text{Animals}} = N_{\text{Dream}} \cap N_{\text{Animals}}$ is the set of animal characters, and $C_{\text{Fictional}} = N_{\text{Dream}} \cap N_{\text{Fictional}}$ is the set of fictional characters. Note that the tool does not use pronouns to identify characters because: (i) the dreamer (most often referred to as 'I' in the

**Table 3.** Excerpts of dream reports with corresponding annotations. (The unique characters in the excerpts are underlined, and our tool's annotations are reported on top of the words in italic.)

| dream text |
|---|
| *male* <u>Roland</u> *aggression, negemo* hit the <u>monster</u> with his sword. The monster *negemo* agonized and *negemo, dead* died |
| I am in hand to hand combat with a *male* <u>man</u> of about fifty. With a *negemo* sinisterleer he *negemo* taunts me. As my knife becomes a fork, I *aggression* spear him in the throat, causing him a *dead* mortalwound His eyes bulge forward; the gushing blood mixes with his last gurgling breath. |
| I dreamed that I *sexual* had sex with *male* <u>Doug</u>. We were in this huge, hot room, and then we just started stripping. He *friendly* caressed me and told me it was the 4^{th} *posemo* happiest moment in his life. Then the rest of the school began to walk in and we started getting dressed really quickly. |
| *female* <u>Bonnie</u> and I are in a room and there is a rattler *animal* <u>snake</u> I am *negemo* afraid of it. Bonnie is unconcerned and picks it up by the tail and I *negemo* cry out in *negemo* fear She *posemo* laughs at me as it wriggles around her and then nearly *aggression* biting me. I move away, *negemo* angry at her for putting me in this *negemo* danger. |

reports) is not considered as a character in the Hall–Van de Castle guidelines; and (ii) our assumption is that dream reports are self-contained, in that, all characters are introduced with a common or proper name.

With these three sets of characters, the tool then computes the first character-related measure among those in table 1:

$$\text{Animals}\% = \frac{|C_{\text{Animals}}|}{|C|}. \tag{4.2}$$

### 4.3.3. Properties of characters

After extracting the characters, the tool classifies them according to two dimensions: sex and 'being dead'.

In line with the official guidelines for dream coding, the tool identifies the sex of *people* characters only, and it does so as follows. If the character is introduced with a common name, the tool searches the character (noun) on Wikidata for the property *sex or gender*. Alternatively, if the character is introduced with a proper name, the tool matches the character with a custom list of 32 055 names whose sex is known—as it is commonly done in gender studies that deal with unstructured text data from the Web [74,75]. In so doing, the tool builds two additional sets from the dream report: the set of male characters $C_{\text{Males}}$, and that of female characters $C_{\text{Females}}$.

To have the tool being able to identify dead characters (who form the set of imaginary characters together with the previously identified fictional characters), we compiled an initial list of death-related words taken from the original guidelines [16,26] (e.g. dead, die, corpse), and manually expanded that list with synonyms from thesaurus.com to increase coverage, which left us with a final list of 20 words.

The tool then matches these terms with all the nodes in the dream report's tree. For each matching node (i.e. for each death-related word), the tool computes the distance between that node and each of the other nodes previously identified as 'characters'. The tool marks the character at the closest distance as 'dead' and adds it to the set of dead characters $C_{\text{Dead}}$. The distance between any two nodes $u$ and $v$ in the tree is calculated with the standard formula:

$$\text{dist}(u, v) = \text{depth}(u) + \text{depth}(u) - 2 * \text{depth}(\text{LCA}(u, v)), \tag{4.3}$$

where depth is the function that computes the depth of a node with respect to the root, and LCA is the lowest common ancestor of the two nodes.

By using the sets of female, male and imaginary (fictional ± dead) characters, the tool computes two additional metrics from table 1:

$$\text{Male}\% = \frac{|C_{\text{Male}}|}{|C_{\text{Male}}| + |C_{\text{Female}}|} \tag{4.4}$$

and

$$\text{Imaginary}\% = \frac{|C_{\text{Dead}}| + |C_{\text{Fictional}}|}{|C|}. \tag{4.5}$$

### 4.3.4. Identification of social interactions

To identify social interactions, we first built a database of verbs expressing aggression, friendliness and sexual contacts. From Wordnet, we collected all the verbs under the lexical domains of *verb.contact* and *verb.communication*. We manually filtered all these sets to produce three resulting verb sets of: aggression verbs $V_{\text{Aggression}}$ (361 words), friendliness verbs $V_{\text{Friendly}}$ (70 words) and sexual interaction verbs $V_{\text{Sexual}}$ (70 words).

Having these three sets, the tool is able to determine which nodes in the tree are interaction verbs. Whenever the tool detects an interaction verb, it marks the noun immediately preceding it and following it as potential interaction actors, matches the pair of nouns to the previous sets of characters, and counts each matched pair as being part of an interaction.[2] Because an interaction is of three types, the tool gathers the previously extracted interactions in the set $I = I_{\text{Aggression}} \cup I_{\text{Friendly}} \cup I_{\text{Sexual}}$, where $I_{\text{Aggression}}$ is the set of aggressive interactions, $I_{\text{Friendly}}$ is the set of friendly interactions and $I_{\text{Sexual}}$ is the set of sexual interactions. With those sets of interaction types, the tools computes five additional metrics from table 1:

$$\frac{F}{C} \text{ Index} = \frac{|I_{\text{Friendly}}|}{|C|}, \tag{4.6}$$

$$\frac{A}{C} \text{ Index} = \frac{|I_{\text{Aggression}}|}{|C|}, \tag{4.7}$$

$$\frac{S}{C} \text{ Index} = \frac{|I_{\text{Sexual}}|}{|C|}, \tag{4.8}$$

$$\text{Aggression}\% = \frac{|I_{\text{Aggression}}|}{|I_{\text{Aggression}}| + |I_{\text{Friendly}}| + |I_{\text{Sexual}}|} \tag{4.9}$$

and

$$\frac{\text{Aggression}}{\text{Friendliness}}\% = \frac{|I_{\text{Aggression}}|}{|I_{\text{Friendly}}|}. \tag{4.10}$$

### 4.3.5. Identification of emotions

To identify positive and negative emotions in the dream report, the tool incorporates the Emolex emotion dictionary [76], a widely used lexicon of English words associated with the eight basic emotions of Plutchik's model [77]: anger, fear, anticipation, trust, surprise, sadness, joy and disgust. The Hall–Van de Castle guidelines define that expressions of *joy* is indicated by the presence of positive emotions, whereas expressions of *anger*, *sadness* and *apprehension* are indicated by markers of negative emotions. The tool therefore compiles a list of emotion words $W_{\text{Emotions}}$ that is composed by positive words $W_{\text{Positive}}$ (containing all the words in the dream report that belong to the set of the Emolex category of *joy*) and by negative words $W_{\text{Negative}}$ (containing all the dream report's words belonging to the Emolex categories of *anger*, *sadness* or *fear*). Overall, Emolex contains 689 emotion terms. In so doing, it is able to compute (table 1):

$$\text{Negemo}\% = \frac{|W_{\text{Negative}}|}{|W_{\text{Positive}}| + |W_{\text{Negative}}|}. \tag{4.11}$$

### 4.3.6. Normalized *h* profiles

As mentioned in §4.1, all the measures need to be normalized using Cohen's *h* (equation (4.1)) against normative scores that express the values found in a 'typical' dream of a healthy individual (the normalized measures for a set of dream reports form what researchers call the set's '*h*-profile'). Traditionally, these values were computed on the *normative set* (§4.2.1) of approximately 1000 hand-coded dreams of European-American students [16]. By taking advantage of our ability to score dream reports at scale, we computed our own norms by scoring all the dream reports in the *no-condition set* and taking the average value for each of metrics in table 1.

---

[2]Note that the tool does not encode an interaction's directionality as the coding requires only to count the number of interaction types.

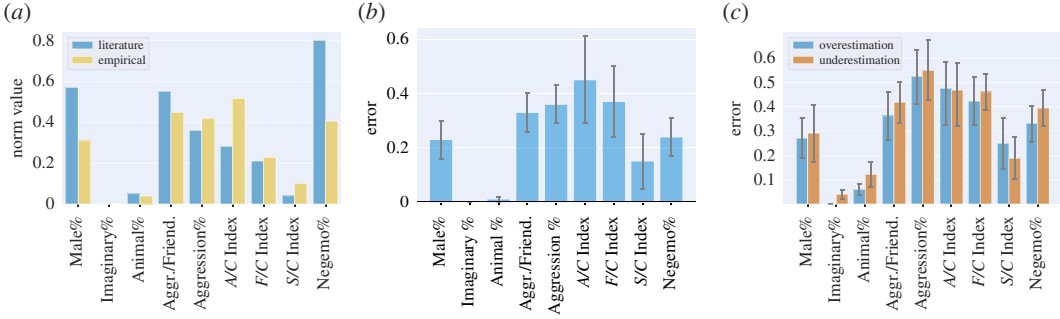

**Figure 3.** Evaluation of our dream scoring tool. (*a*) Differences between the norms historically used in the literature (reflecting a student population) and the norms empirically computed on a large sample of dream reports (reflecting the US population). (*b*) The average errors of our tool's measurements computed on a hand-coded set of dream reports. (*c*) Errors of overestimation and errors of underestimation—zero-error instances are not considered.

There are two main advantages in using these empirically computed norms. First, given the difficulty of annotating a large number of dream reports, the norms historically used in the literature were computed on a limited set that only included students (the 'normative set' in this paper). By contrast, being automatically computed on the majority of our dream reports (those in the 'no-condition set'), the empirical norms reflect the larger United States (US) population. Second, by using the empirical norms, any systematic overestimation or underestimation done by the tool is compensated, as the empirical norms come from the very same tool.

Based on figure 3*a*, we can see that the empirical norms are close to the traditional ones for the measures of Imaginary%, $S/C$ Index, $F/C$ Index, Aggression% and Aggression/Friendliness. The larger gap for Male%, Animal%, Negemo% is probably explained by an over-representation of these concepts in the *normative set*, while the gap for the $A/C$ Index is larger and probably comes from a combination of the topical skewness in the *normative set* and our tool's misclassification, which we will explore next.

## 4.4. Evaluation

We evaluated our tool using two sets of dream reports that have been hand-coded by dream experts using the Hall–Van de Castle system (§4.2.1): (i) the *annotated set* of dream reports, and (ii) the *normative set* from which the norms used in the literature were computed. For all those dream reports, we measured the extent to which the sets of characters, interaction and emotions estimated by the dream processing tool matched the corresponding ground-truth sets; table 4 summarizes the resulting precision, recall and F1-score.

We then proceeded to compare the the Hall–Van de Castle indicators computed by our tool (table 1) with the corresponding ground-truth values. Given the ground-truth value $v$ and the tool's value $\overline{v}$, we computed the error as $e = |v - \overline{v}|$.

Overall, the average error across categories is 0.24 (figure 3*b*), which is limited considering the high variability of textual styles in the corpus, and the inherent complexity of some of the measures. To interpret the magnitude of the error, one should consider that, in practice, all the indicators take on values that are almost always in the [0,1] range on this specific test set of dream reports. The measure that deviates most from this range is the $A/C$ Index: it is greater than 1 in 6% of the cases in the ground-truth and in 3% of the cases according to our tool. The $A/C$ Index, is also affected by the highest error ($e = 0.45$). This is partly because its range is slightly higher than those of other indicators, and because it requires the identification of characters and the detection of acts of aggression, which are potentially ambiguous in their interpretation and, as such, are hard to be automatically extracted. As we have previously mentioned, to partly mitigate the impact of the tool's errors on the computation of *h*-profiles, we normalized all our metrics using the empirically defined norms. In our corpus, unlike aggression acts which tend to take a variety of forms, sexual interactions take predictable forms, typically involve two individuals having sex, and, as such, are easier to automatically identify; friendly interactions, on the other hand, are identified with a level of difficulty that is between aggression acts' and friendly interactions'.

In addition to reporting absolute errors, we separately report errors of overestimation ($e_{over} = v - \overline{v}$ if $v - \overline{v} > 0$) and of underestimation ($e_{under} = |v - \overline{v}|$ if $v - \overline{v} < 0$), which are computed without considering zero-error instances (figure 3*c*). Overall, each pair of bars are aligned; the more

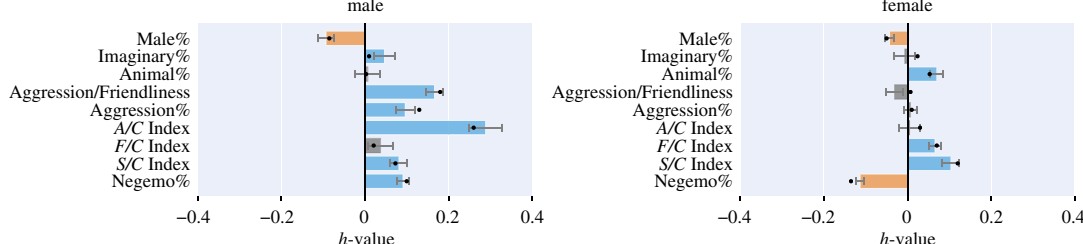

**Figure 4.** The $h$-profile computed on the male reports and that on the female reports. The more $h$ for a set (e.g. male set) of dream reports deviates from 0, the more the reports deviate from those of healthy individuals. The black bullets represent the $h$-values by macro-averaging the averages computed for dream reports during each decade separately. The bars corresponding to values that do not differ significantly from the norm ($p > 0.01$) are greyed-out.

**Table 4.** Precision, recall and F1-score for different sets of elements extracted by the dream processing tool against the hand-coded sets.

| indicators | precision | recall | F1 |
|---|---|---|---|
| all characters ($C$) | 0.71 | 0.98 | 0.75 |
| male ($C_{Male}$) | 0.58 | 0.58 | 0.58 |
| female ($C_{Female}$) | 0.59 | 0.61 | 0.58 |
| animals ($C_{Animals}$) | 0.40 | 0.40 | 0.40 |
| dead and imaginary ($C_{Imaginary}$) | 0.22 | 0.22 | 0.22 |
| friendly ($I_{Friendly}$) | 0.34 | 0.43 | 0.36 |
| aggressive ($I_{Aggressive}$) | 0.33 | 0.48 | 0.36 |
| sexual ($I_{Sexual}$) | 0.16 | 0.17 | 0.17 |
| emotions ($W_{Emotions}$) | 0.23 | 0.35 | 0.27 |

aligned each pair of bars, the better. That is because alignment indicates that overestimation is comparable to underestimation and, in a large set, their effects partly cancel themselves out and, as such, end up having little impact on our results.

# 5. Testing the five research hypotheses

After having ascertained the validity of our tool's output and applying it to the sets of dream reports described in §4.2.1, we set out to test our five hypotheses.

**H1** Female's dreams are characterized by emotions rather than interactions around activities, and by limited levels of aggression.

Male and female dream reports differ on a number of key aspects. As opposed to female reports, *male* ones contained more aggression markers and, as a result, more negative emotions (figure 4). The $A/C$ Index is particularly high ($h > 0.2$). Although this index might be overestimated by our tool, the correction applied by the empirical norms ensures that male dream reports contain a large number of acts of aggression. By contrast, female reports contained more positive emotions and more friendly interactions, which is in line with our first hypothesis.

The number of male and female dream reports varied across the decades in a non-uniform fashion (figure 1). To check whether the trends observed were actually associated with gender variation and not confounded by the decade, we calculated the $h$-profiles a second time, using a different approach. We first grouped dream reports by decade (starting from the 1930s, as earlier decades are too sparse), computed the per-decade averages and then macro-averaged across decades. By doing so, we weighted every decade equally, regardless of their dream count, thus mitigating imbalance biases. For the sake of comparison, we overlay the newly calculated macro-averages as bullets on top of the figure 4 histogram.

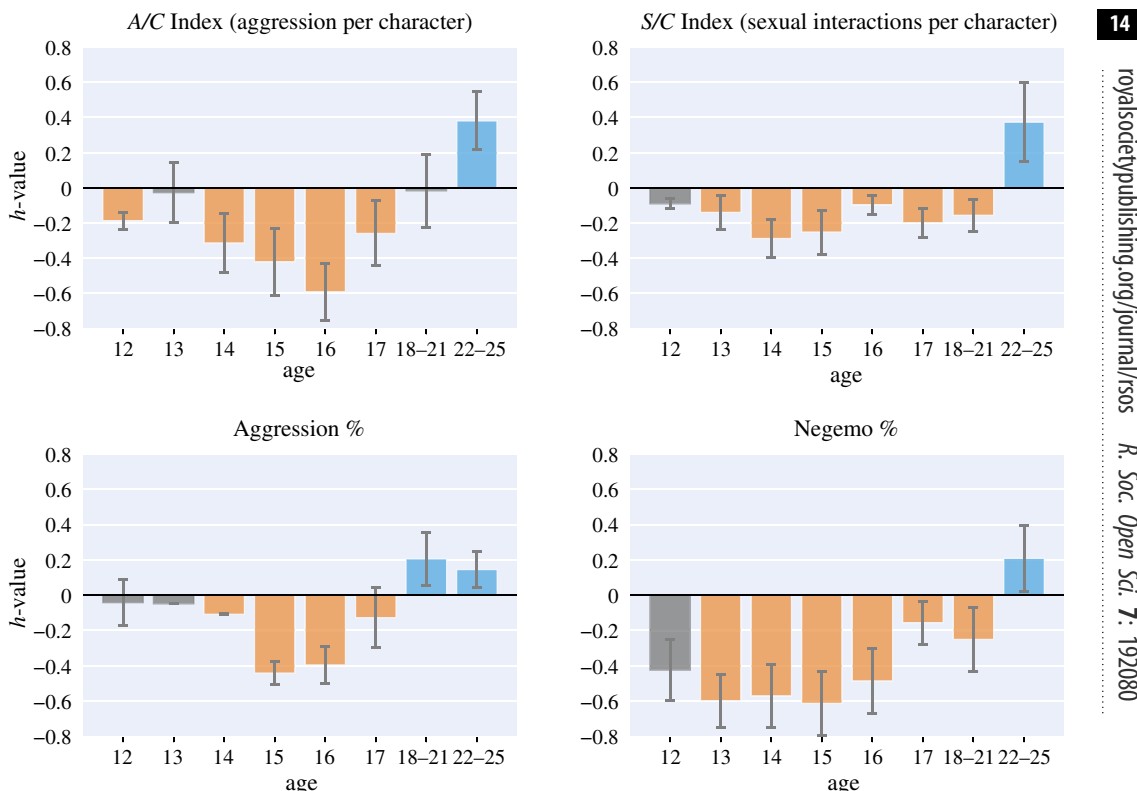

**Figure 5.** The *h*-profiles computed on Izzy's dream reports. Izzy is a young woman who systematically documented her dreams from the age of 12 to the age of 25. The more *h* for a set of reports deviates from 0, the more the reports deviate from those of healthy individuals. The bars corresponding to the values that do not differ significantly from the norm ($p > 0.01$) are greyed-out.

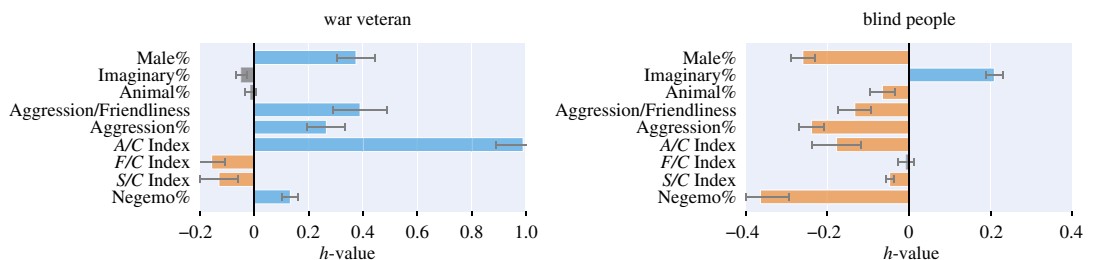

**Figure 6.** The *h*-profiles computed on the war veterans' reports and on the blind people's. The more *h* for a set of reports deviates from 0, the more the reports deviate from those individuals with no declared condition. The bars corresponding to the values that do not differ significantly from the norm ($p > 0.01$) are greyed-out.

The trends obtained were consistent across the two calculations. In our sets of data, because the dreamer's age is often unknown, we were not able to estimate the interaction effects between age and sex.

**H2** An adolescent's dreams are characterized by negative emotions and aggression, followed by sexual interactions in early adult life.

In the case of our prolific dreamer Izzy, the presence of negative emotions and aggression in her dream reports increased during adolescence, and sexual interactions appeared after the age of 18–21 (figure 5), which is in line with our hypothesis. Given that these results are based on one dreamer only, we cannot say that our hypothesis has been verified. However, Izzy is one of the few young dreamers in our data, and the only one reporting an extensive set of dreams at different stages of her life. To further test our second hypothesis, future work should focus on collecting dream reports of adolescents at a larger scale.

**H3** A war veteran's dreams are characterized by negative emotions and aggression.
**H4** Blind people's dreams feature more imaginary characters and aspects related to their real-life carers (e.g. they feature women).

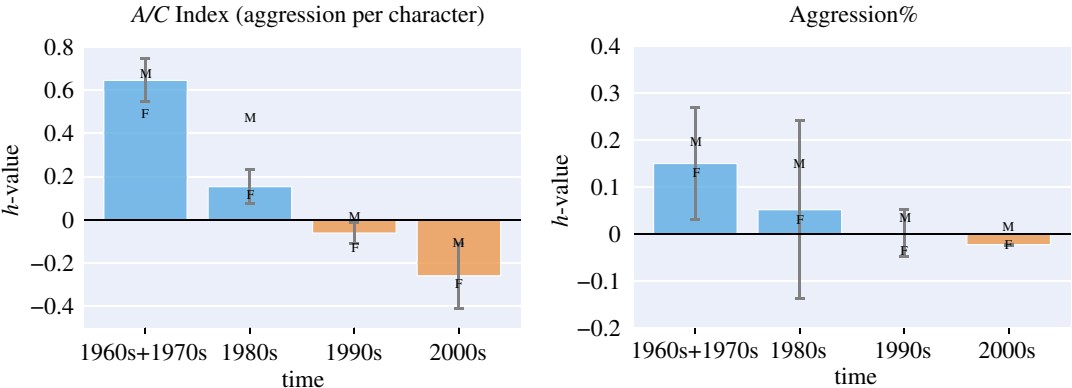

**Figure 7.** The *h*-profile of $A/C$ Index and of Aggression% computed on the dream reports at each decade, from 1960 to 2000. The more *h* for a set of reports (e.g. reports in the 1960s) deviates from 0, the more the reports deviate from those individuals with no declared condition. The M and F markers indicate the values computed on dream reports exclusively from male or female dreamers.

The *war veteran*'s reports contained aggression, more male characters and less sexual interactions than the reports in the *no-condition set* (figure 6), which matches our third hypothesis. Again, given that these results are based on one dreamer only, we cannot say that our hypothesis has been verified. To test our third hypothesis further, future work should focus on collecting dream reports from a variety of individuals suffering from post-traumatic stress disorder at a larger scale.

In the same figure, we report the results for blind people. Their reports tended to contain more female and imaginary characters, and that is in line with our hypothesis. Furthermore, these reports contained considerably fewer episodes of aggression and reduced prevalence of negative emotions. These findings have not been reported in previous research but might be partly explained by the over-representation of middle-age individuals (that is, more emotional stable individuals [78,79]) in the blind set.

**H5** Dreams during times of societal aggression (in the 1960s) are characterized by aggression.

Among all dreamers, from 1960 to 2000, the level of aggression in dream reports was highest in the 1960s and then steadily decreased (figure 7), which matches official statistics of violent crimes in the USA [57].

Similar to what we have done to verify the gender hypothesis, to check if our findings were affected by the uneven distribution of males and females across decades, we re-computed the *h*-values of different decades by considering only male or only female dreamers. The new results are shown on top of the histogram in figure 7 with small M and F markers. Aggression for both men and women steadily decreased over the decades, even though the male values are always higher than those measured from female's dream reports.

# 6. Discussion and conclusion

## 6.1. Limitations

This study has several limitations that call for further investigation in the future.

### 6.1.1. Demographic bias

Our dreamers are mainly well-educated individuals living in the USA. Despite that, our samples allowed us to segment reports in such a way that made it possible to individually study differences in gender, age, life-changing experiences and even impact of societal aggression.

### 6.1.2. Data limitations

Our sets of data have two main limitations. The first has to do with data segmentation: when studying individual characteristics that were not prevalent in our sets, the number of dreamers was low (e.g. our results on adolescence and on post-traumatic stress disorder come from the analysis of hundreds of dream reports, but those reports were recorded by only two individuals). The limited number of those

who reported their dreams made it possible to only find supporting rather than conclusive evidence for the continuity hypothesis. Additionally, many dream reports are associated with groups of people (e.g. young male students, brides) rather than being linked to individual identities, which makes it hard to compute any individual-level $h$-profile. The second limitation has to do with the dreamer profiles: important individual characteristics of the dreamers were missing. As such, we could not account for interaction effects: for example, when assessing sex differences, we could not control for age as it was not available. Also, this lack of data prevented us making an assessment of the set of dreamers considered being a representative sample of the population in USA. To partly counter these limitations, researchers should collect sets of reports that come with rich descriptions about the dreamers, that reflect a variety of segments of the general population, and that are collected over a long period of time.

### 6.1.3. Accuracy of natural language processing tools

The main objective of this work was to test the continuity hypothesis at scale, and that entailed being able to automatically analyse the content of dream reports. In the future, for specific content categories (e.g. acts of aggressions, friendly interactions), classification could be improved by conducting further research on NLP tools that not only are able to extract textual descriptors from a sentence but also are able to characterize the contexts of those descriptors while dealing with the nuances of the human language.

### 6.1.4. Causality

Our results do not speak to causality. One way to counter that in the future would be to run controlled studies in which dream reports and waking life experiences are collected together for a sufficiently large number of individuals over a long period of time.

### 6.1.5. Recollection bias

A final limitation is that we have studied not what our individuals dreamed but what they remembered to have dreamed. We analysed dream reports, which probably represent only part of the dreams our individuals had. Some dreams are forgotten, others are remembered but not reported, while others are reported in narrative forms. Given the reports' narrative structures, individuals tend to selectively express only the most salient experiences [31].

## 6.2. Implications

### 6.2.1. Theoretical implications

Our main theoretical contribution was to test the continuity hypothesis at scale. We found supporting evidence. Dream reports contained statistical markers that reflected what the dreamers probably experienced in real life, and they did so in expected ways. In her transition from adolescence to womanhood, Izzy started to colour her dreams with sexual images (her sexual imaginary's deviation from adult norms went from $h \approx -0.2$ in her teenage years to $h \approx 0.4$ in her 20s). Even more strikingly, in the case of the war veteran's dreams, the frequency of aggression acts by far exceeded that of a typical dreamer, reaching an $h$ as high as $\approx 0.9$. More generally, sleep scientists have speculated that 'dream content may provide us with different information about people than most personality tests do' [26, p. 157]. With our tool at hand, researchers could test the extent to which that holds at scale.

### 6.2.2. Practical implications

Our scoring tool makes it possible to build technologies that require automatic classification of dream reports. In the future, this could result in automatic diagnostic or prognostic indicators for mental health in general. We have also shown that different levels of violent crime in a society were reflected in dream reports. Given that, being able to automatically analysing dream reports could result in well-being tools for communities or even nations.

## 6.3. Conclusion

By going beyond manual annotation of dream reports, we were able to study the relationship between well-being and dreaming, which has been largely unknown in non-clinical populations. We found that most dream reports were indeed a continuation of what our dreamers were likely to experience in real life. As much as in their real lives, in their dreams, women tended to be friendlier and less aggressive than men; Izzy experienced negative emotions during her adolescence, followed by sexual interactions in later years; our war veteran experienced uncommon levels of aggression; and individuals in the US experienced high levels of aggression in the 1960s, in line with official crime statistics. All these results support the idea that there is indeed continuity between what individuals experience in real life and what they dream. Interestingly, we found that blind people—who were expected to dream in ways similar to the general public—tended to instead dream imaginary characters.

In the future, we will explore which research communities and practitioners could benefit from our dream scoring tool. We will also integrate our tool with a mobile app with which users can record their dreams in a convenient way. The app will also collect self-reported well-being scores, making it possible to test the causal relationship between well-being and dreaming. On the consumer side, our methods—which make our 'sleeping mind' quantifiable—might well help designers rethink current quantified-self technologies.

Data accessibility. The original dream report data are available at http://dreambank.net. The scores assigned by our tool to the dream reports we analysed are available at https://doi.org/10.5061/dryad.qbzkh18fr [80]. The data and code to extract the Hall–Van de Castle's dream coding scale is also available at http://social-dynamics.net/dreams.

Authors' contributions. A.F. collected the data and conducted the experiments. L.M.A. and D.Q. conceived the experiments and wrote the manuscript.

Competing interests. We declare we have no competing interests.

Funding. We received no funding for this study.

Acknowledgements. We thank Prof. William Domhoff for providing the dream reports data.

# Appendix A

The dream sets we considered in our analysis are composed by multiple groups of dream reports from Dreambank. Table 5 lists the Dreambank groups corresponding to each of the sets. Their description is available at http://www.dreambank.net/grid.cgi. [80]

**Table 5.** Group of dream reports from Dreambank included in our dream sets.

| dream set | dreambank groups |
| --- | --- |
| full set | union of all groups below |
| no-condition set | union of all groups below except: blind-f, blind-m, edna, vietnam_vet, vietnam_vet2, vietnam_vet3 |
| male set | bosnak, chuck, david, ed, emmas_husband, jeff, kenneth, lawrence, mack, mark, melvin, midwest_teens-m, natural_scientist, pegasus, peru-m, phil1, phil2, phil3, physiologist, ringo, toby, tom |
| female set | alta, angie, arlie, b, b2, bay_area_girls_456, bay_area_girls_789, bea1, bea2, dahlia, dorothea, elizabeth, emma, esther, hall_female, izzy, jasmine1, jasmine2, jasmine3, jasmine4, madeline1-hs, madeline2-dorms, madeline3-offcampus, madeline4-postgrad, melissa, melora, merri, midwest_teens-f, nancy, peru-f, samantha, seventh_graders, ucsc_women, vickie, wedding, west_coast_teens |
| blindness set | blind-f, blind-m, edna |
| war veteran set | vietnam_vet, vietnam_vet2, vietnam_vet3 |
| Izzy set | izzy12, izzy13, izzy14, izzy15, izzy16, izzy17, izzy18, izzy22 |
| normative set | norms-f, norms-m |

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
