## [Reviewer comments · Royal Society Open Science]

Review History

RSOS-192080.R0 (Original submission)

Review form: Reviewer 1

Is the manuscript scientifically sound in its present form?

Yes

Are the interpretations and conclusions justified by the results?

Yes

Is the language acceptable?

Yes

Do you have any ethical concerns with this paper?

No

Have you any concerns about statistical analyses in this paper?

No

Recommendation?

Accept with minor revision (please list in comments)

Comments to the Author(s)

I thank the authors for the interesting read. Yet, I have a few concerns that I hope the authors will clarify.

First, computing h scores with respect to dataset-specific averages (section 4 c vi) comes with a few assumptions and limitations. There is limited information on to what extent the dataset is representative of the us population, comprised of emotionally stable adults, or psychologically healthy individuals, which are assumptions guiding the interpretation of the h values in section 5. Furthermore, although this process allows for relative comparisons within it, it somewhat hinders robustness with respect to new data from other domains. I am not fully convinced that switching the reference normative set is an advantage for the tool in general, beyond proving the research hypotheses of the paper.

Second, the authors do not give information on the skew of individual dreamers' proficiency in the dataset, and do not account for individual differences in the analyses. Controlling for individual differences would better fit the analyses, since ultimately the continuity hypothesis applies to individuals, and if dreamers like Izzy dominate the datasets it would be hard to interpret the findings.

I overall found the paper engaging and the results fascinating, and would consider it for publication given that the above concerns are addressed.

Review form: Reviewer 2

Is the manuscript scientifically sound in its present form?

No

Are the interpretations and conclusions justified by the results?

No

Is the language acceptable?

Yes

Do you have any ethical concerns with this paper?

No

Have you any concerns about statistical analyses in this paper?

Yes

Recommendation?

Major revision is needed (please make suggestions in comments)

Comments to the Author(s)

Summary

The authors present an automated method for analyzing dream reports by operationalizing Hall & Van de Castle's dream analysis scale. They validate their tool using 1700 manually annotated dream reports. They then apply their method to analyse 24 K dream reports and they examine whether continuity hypothesis holds.

Reasons to accept

- Broadly speaking, utilizing of NLP techniques to assist in manually intensive tasks in non-computational fields of study could help to scale up the size of studies typically performed and the authors present such an application of NLP to analyze dream reports.
- I like the simplified research hypotheses that the authors have formulated to find evidence for 'continuity hypothesis'.

Reasons to reject

- From the title of the paper it seems that the method (automatic analysis of dream reports) is the main goal. However, I find the method description and evaluation lacking details necessary for me to understand them fully.
- Moreover, the authors also claim that the "the main objective of this work was to test the 'continuity hypothesis' at scale" (page 17, line 48). I find that the datasets that the authors utilize to validate their hypotheses to test continuity hypothesis, to be too small in certain cases to make conclusive inferences (as the authors also point out). Additionally, I am not sure if the effects of confounding variables (for instance age or decade while examining hypothesis 1 or gender distribution for different decades while examining hypothesis 5) have been sufficiently disentangled or controlled for. Therefore I am unsure if the authors have succeeded in the main objective of their work.

Specific concerns and points for improvement

In my opinion there are two main aims of this paper and I next point out my concerns and suggestions for improvement regarding these two goals:

Method: To present an automated method for operationalizing Hall & Van de Castle's dream analysis scale.

- I like the succinct description of the Hall and Van de castle's coding system that the authors provide on page 4, lines 16-20. The authors also mention on page 6 that the coding system consists of ten categories and sub categories. It would be great if the authors could give a brief description of the coding system and also explain why they chose to only code the dream reports along their selected three categories. Could the exclusion of the other categories hinder the inferences that they can make. I believe such a description would make the method section more complete and make it easier for non-experts to also follow their paper.

- I like Table 1, it gives a very good overview of what the tool measures.

- On page 7, lines 48-51, authors indicate that they also performed the paired t-tests. I do not see where they have reported these results in the manuscript.

- It is not immediately clear to me which dreams on dreambank.net belong to which of author's 7 datasets. It would greatly help reproducibility if the authors would provide this information. Perhaps the authors have included it at <https://doi.org/10.5061/dryad.n02v6wwsq>, however I could not access it (DOI not found error) to verify it.

- I would have also liked to see an evaluation at each step of the automation pipeline to better understand which of the automation steps work well and which don't. This is especially important since different dictionary approaches and NLP methods have been employed to infer the different measured quantities and its not clear that they all work equally well.

- For instance, since the authors identify the different characters (C_people, C_animals, and C_fictional), and the normative set contains these manual annotations for the characters in about 1000 dream reports. Authors could present the precision and recall numbers for the different types of characters.

- Similar method could be used to measure and report the precision and recall using the normative set for other measured quantities like C_males, C_females, C_dead, I_friendly, I_aggression, I_sexual, W_negative and W_positive. A table summarizing these results would help future researchers to estimate how well the tool would work for other related scenarios for dream analysis.

- It would be great if the authors could also report the size of the final lists they used for identifying characters, character's properties, social interactions and emotions.
- Why does the size of the normative set differ when reported in different parts of the paper -- 1000 on page 9, 600 on page 13.
- I like that the authors generate the normalized h profiles using the scores given out by their tool for the no condition dataset. I agree that this would help mitigate some of the errors of their tool as well as capture a wider distribution of population.
- I would encourage the authors to take a bit more care in explaining their evaluation. While Figure 3 is quite informative, I think it is not sufficiently explained in the text (especially Figure 3A). A bit more explanation would help the readers to follow along more easily.
- I also wonder what the errors look like for the normative dataset. It would be great if they authors could include the normative set results (which they call literature) in Figure 3B and 3C to help others contextualize their evaluation results. Also, please use consistent terminology and dataset names.

Application: To utilize this automated method to analyze a large set of dream reports to test the continuity hypothesis.

- The results for 2 of their 5 hypotheses regarding adolescence (1 dreamer) and post war trauma (1 dreamer) are based on the dreams of a single dreamer and as the authors themselves point out, this makes it hard to verify these hypotheses with confidence.
- For another hypothesis regarding blind people (16 dreamers), again the results are based on 16 dreamers (though a higher number of dreams), and I am not sure if this number is sufficient to consider the hypothesis verified.
- Page 15, line 39, authors mention that "based on available descriptions about the dreamers, the distribution of ages in the two sets should be comparable...". I would be more convinced if the authors could include what information available in the descriptions convinced them of the similar age distributions.
- Additionally I noticed that the male and female datasets include dreamers from a wide range of decades 1910 - 2010. Are the reports from males and females equally distributed across the decades, otherwise the difference in the decades might also lead to the differences in the observed level of aggression etc.
- For the annotated set, in addition to reporting the number of dreams in each decade and the gender of the dreamers, it would be helpful to also report the number of dreamers for each decade. It is again unclear to me, how many people's dreams are utilized to test hypothesis 5. Also if the gender distribution similar for the different decades? Otherwise the increased amount of aggression for 1960's might just be a consequence of mostly men's dreams having being analyzed.
- In general, I am not sure if the effects of confounding variables (for instance age or decade while examining hypothesis 1 or gender distribution for different decades while examining hypothesis 5) have been sufficiently disentangled or controlled for. Perhaps more sophisticated statistical testing techniques would be useful for gaining more confidence in the results.
- For the hypothesis 5, how did the other aggressive interaction features (aggression% and aggression/friendliness) vary with the decades.

Decision letter (RSOS-192080.R0)

23-Mar-2020

Dear Dr Aiello,

The editors assigned to your paper ("Our Dreams, Our Selves: Automatic Analysis of Dream Reports") have now received comments from reviewers. We would like you to revise your paper in accordance with the referee and Associate Editor suggestions which can be found below (not

including confidential reports to the Editor). Please note this decision does not guarantee eventual acceptance.

Please submit a copy of your revised paper before 15-Apr-2020. Please note that the revision deadline will expire at 00.00am on this date. If we do not hear from you within this time then it will be assumed that the paper has been withdrawn. In exceptional circumstances, extensions may be possible if agreed with the Editorial Office in advance. We do not allow multiple rounds of revision so we urge you to make every effort to fully address all of the comments at this stage. If deemed necessary by the Editors, your manuscript will be sent back to one or more of the original reviewers for assessment. If the original reviewers are not available, we may invite new reviewers.

- Data accessibility

If you wish to submit your supporting data or code to Dryad (<http://datadryad.org/>), or modify your current submission to dryad, please use the following link:
<http://datadryad.org/submit?journalID=RSOS&manu=RSOS-192080>

- Competing interests

- Authors' contributions

All submissions, other than those with a single author, must include an Authors' Contributions section which individually lists the specific contribution of each author. The list of Authors should meet all of the following criteria; 1) substantial contributions to conception and design, or

acquisition of data, or analysis and interpretation of data; 2) drafting the article or revising it critically for important intellectual content; and 3) final approval of the version to be published.

- Acknowledgements

- Funding statement

Kind regards,
Andrew Dunn
Senior Publishing Editor
Royal Society Open Science Editorial Office
Royal Society Open Science
openscience@royalsociety.org

on behalf of Professor Mirella Lapata (Associate Editor) and Marta Kwiatkowska (Subject Editor)
openscience@royalsociety.org

Associate Editor's comments (Professor Mirella Lapata):

The manuscript is very interesting and we would like it to appear in the journal. The second reviewer has several questions/suggestions which if addressed would make the manuscript stronger. We are looking forward to receiving the revised version which should address the following:

- 1) Give a more detailed description on the coding system
- 2) Clarify issues with provenance of the dataset, providing statistics for age, sex, etc.
- 3) Evaluate subcomponents of proposed method and in general fix issues with evaluation raised by both reviewers (e.g., remove confounds in analysis).

Comments to Author:

Reviewers' Comments to Author:

Reviewer: 1

Comments to the Author(s)

I thank the authors for the interesting read. Yet, I have a few concerns that I hope the authors will clarify.

First, computing h scores with respect to dataset-specific averages (section 4 c vi) comes with a few assumptions and limitations. There is limited information on to what extent the dataset is representative of the us population, comprised of emotionally stable adults, or psychologically healthy individuals, which are assumptions guiding the interpretation of the h values in section 5. Furthermore, although this process allows for relative comparisons within it, it somewhat hinders robustness with respect to new data from other domains. I am not fully convinced that switching the reference normative set is an advantage for the tool in general, beyond proving the research hypotheses of the paper.

Second, the authors do not give information on the skew of individual dreamers' proficiency in the dataset, and do not account for individual differences in the analyses. Controlling for individual differences would better fit the analyses, since ultimately the continuity hypothesis applies to individuals, and if dreamers like Izzy dominate the datasets it would be hard to interpret the findings.

I overall found the paper engaging and the results fascinating, and would consider it for publication given that the above concerns are addressed.

Reviewer: 2

Comments to the Author(s)

Summary

The authors present an automated method for analyzing dream reports by operationalizing Hall & Van de Castle's dream analysis scale. They validate their tool using 1700 manually annotated dream reports. They then apply their method to analyse 24 K dream reports and they examine whether continuity hypothesis holds.

Reasons to accept

- Broadly speaking, utilizing of NLP techniques to assist in manually intensive tasks in non-computational fields of study could help to scale up the size of studies typically performed and the authors present such an application of NLP to analyze dream reports.
- I like the simplified research hypotheses that the authors have formulated to find evidence for 'continuity hypothesis'.

Reasons to reject

- From the title of the paper it seems that the method (automatic analysis of dream reports) is the main goal. However, I find the method description and evaluation lacking details necessary for me to understand them fully.
- Moreover, the authors also claim that the " the main objective of this work was to test the 'continuity hypothesis' at scale" (page 17, line 48). I find that the datasets that the authors utilize to validate their hypotheses to test continuity hypothesis, to be too small in certain cases to make conclusive inferences (as the authors also point out). Additionally, I am not sure if the effects of confounding variables (for instance age or decade while examining hypothesis 1 or gender distribution for different decades while examining hypothesis 5) have been sufficiently disentangled or controlled for. Therefore I am unsure if the authors have succeeded in the main objective of their work.

Specific concerns and points for improvement

In my opinion there are two main aims of this paper and I next point out my concerns and suggestions for improvement regarding these two goals:

Method: To present an automated method for operationalizing Hall & Van de Castle's dream analysis scale.

- I like the succinct description of the Hall and Van de castle's coding system that the authors provide on page 4, lines 16-20. The authors also mention on page 6 that the coding system consists of ten categories and sub categories. It would be great if the authors could give a brief description of the coding system and also explain why they chose to only code the dream reports along their selected three categories. Could the exclusion of the other categories hinder the inferences that they can make. I believe such a description would make the method section more complete and make it easier for non-experts to also follow their paper.

- I like Table 1, it gives a very good overview of what the tool measures.

- On page 7, lines 48-51, authors indicate that they also performed the paired t-tests. I do not see where they have reported these results in the manuscript.

- It is not immediately clear to me which dreams on dreambank.net belong to which of author's 7 datasets. It would greatly help reproducibility if the authors would provide this information. Perhaps the authors have included it at <https://doi.org/10.5061/dryad.n02v6wwsq>, however I could not access it (DOI not found error) to verify it.

- I would have also liked to see an evaluation at each step of the automation pipeline to better understand which of the automation steps work well and which don't. This is especially important since different dictionary approaches and NLP methods have been employed to infer the different measured quantities and its not clear that they all work equally well.

- For instance, since the authors identify the different characters (C_people, C_animals, and C_fictional), and the normative set contains these manual annotations for the characters in about 1000 dream reports. Authors could present the precision and recall numbers for the different types of characters.

- Similar method could be used to measure and report the precision and recall using the normative set for other measured quantities like C_males, C_females, C_dead, I_friendly, I_aggression, I_sexual, W_negative and W_positive. A table summarizing these results would help future researchers to estimate how well the tool would work for other related scenarios for dream analysis.

- It would be great if the authors could also report the size of the final lists they used for identifying characters, character's properties, social interactions and emotions.

- Why does the size of the normative set differ when reported in different parts of the paper -- 1000 on page 9, 600 on page 13.

- I like that the authors generate the normalized h profiles using the scores given out by their tool for the no condition dataset. I agree that this would help mitigate some of the errors of their tool as well as capture a wider distribution of population.

- I would encourage the authors to take a bit more care in explaining their evaluation. While Figure 3 is quite informative, I think it is not sufficiently explained in the text (especially Figure 3A). A bit more explanation would help the readers to follow along more easily.

- I also wonder what the errors look like for the normative dataset. It would be great if they authors could include the normative set results (which they call literature) in Figure 3B and 3C to help others contextualize their evaluation results. Also, please use consistent terminology and dataset names.

Application: To utilize this automated method to analyze a large set of dream reports to test the continuity hypothesis.

- The results for 2 of their 5 hypotheses regarding adolescence (1 dreamer) and post war trauma (1 dreamer) are based on the dreams of a single dreamer and as the authors themselves point out, this makes it hard to verify these hypotheses with confidence.

- For another hypothesis regarding blind people (16 dreamers), again the results are based on 16 dreamers (though a higher number of dreams), and I am not sure if this number is sufficient to consider the hypothesis verified.

- Page 15, line 39, authors mention that "based on available descriptions about the dreamers, the distribution of ages in the two sets should be comparable...". I would be more convinced if the authors could include what information available in the descriptions convinced them of the similar age distributions.

- Additionally I noticed that the male and female datasets include dreamers from a wide range of decades 1910 - 2010. Are the reports from males and females equally distributed across the decades, otherwise the difference in the decades might also lead to the differences in the observed level of aggression etc.

- For the annotated set, in addition to reporting the number of dreams in each decade and the gender of the dreamers, it would be helpful to also report the number of dreamers for each decade. It is again unclear to me, how many people's dreams are utilized to test hypothesis 5. Also if the gender distribution similar for the different decades? Otherwise the increased amount of aggression for 1960's might just be a consequence of mostly men's dreams having being analyzed.

- In general, I am not sure if the effects of confounding variables (for instance age or decade while examining hypothesis 1 or gender distribution for different decades while examining hypothesis 5) have been sufficiently disentangled or controlled for. Perhaps more sophisticated statistical testing techniques would be useful for gaining more confidence in the results.

- For the hypothesis 5, how did the other aggressive interaction features (aggression% and aggression/friendliness) vary with the decades.

Author's Response to Decision Letter for (RSOS-192080.R0)

See Appendix A.

RSOS-192080.R1 (Revision)

Review form: Reviewer 1

Is the manuscript scientifically sound in its present form?

Yes

Are the interpretations and conclusions justified by the results?

Yes

Is the language acceptable?

Yes

Do you have any ethical concerns with this paper?

No

Have you any concerns about statistical analyses in this paper?

No

Recommendation?

Accept with minor revision (please list in comments)

Comments to the Author(s)

The authors significantly improved the clarity of the manuscript with respect to the methods. Their response letter also greatly helped my understanding of the inherent limitations of the data,

and especially the necessity of performing aggregate, rather individual-level analyses. I see the limitations adequately reflected in the manuscript, and I believe that the main contributions of the work stand even in the face of such limitations, given the generality of the proposed framework.

Given the access to the data and the extended clarification of the measures, a minor point became less clear to me, specifically in how errors are computed in figure 3 and sec 4d. To my understanding, the absolute error to validate the model is computed on the raw values for each measure. If this is the case, I am not sure that errors lay on a $[0, 1]$ scale, as reported in the abstract, even for ratio features like Agg./Friend. If I understood correctly, and if no normalization of the raw values is performed, the wider error bars for fig 3b-c may be due to unequal scales or variances in the measures, rather than measurement errors of the tool.

I would also benefit from a slightly more detailed description of table 4. Are performance metrics computed in terms of exact match with the ground truth, e.g. if the number of all characters inferred matches with that of the ground truth, regardless of the magnitude of the discrepancy?

In all other respects, I find the paper improved, therefore I confirm my previous positive judgment.

Review form: Reviewer 3

Is the manuscript scientifically sound in its present form?

Yes

Are the interpretations and conclusions justified by the results?

Yes

Is the language acceptable?

Yes

Do you have any ethical concerns with this paper?

No

Have you any concerns about statistical analyses in this paper?

No

Recommendation?

Accept with minor revision (please list in comments)

Comments to the Author(s)

In the paper titled "Our Dreams, Our Selves: Automatic Analysis of Dream Reports", the authors have designed a NLP tool to score dream reports in Hall & de Castle's scale. The tool identifies proper nouns for characters, and verbs to for interactions between them. I find the paper very interesting, and the accompanying website <http://social-dynamics.net/dreams/> visually beautiful. I also checked the authors' responses to earlier reviews and could see the additions made to the paper. Overall, I think the paper is ready for publication. I've a few minor comments which I would like the authors to consider while preparing the camera ready version.

1. They should discuss a little about the continuity hypothesis in the introduction. It presently appears without the underlying context.

2. I thoroughly enjoyed the background section, but felt that it can be better organized with subsection titles.

3. After presenting necessary background, can all five hypotheses come together?

4. Merge sections 6 and 7 into a unified "Concluding Discussion" section.

5. The xtick labels in Figure 3 are difficult to see. Same is true for the greyed words and legend in Figure 2. Repeating "The bars corresponding to the values that do not differ significantly from the norm ($p > 0.01$) are grayed-out" in every caption doesn't look good.

6. I would request the authors to give a careful pass end-to-end. The grammatical errors need to be fixed. For example: "and they not captured other aspects" (page 3, line 31).

Finally, I enjoyed reading the paper and think that it makes important contribution. Hence, I recommend acceptance of the paper.

Decision letter (RSOS-192080.R1)

Dear Dr Aiello:

On behalf of the Editors, I am pleased to inform you that your Manuscript RSOS-192080.R1 entitled "Our Dreams, Our Selves: Automatic Analysis of Dream Reports" has been accepted for publication in Royal Society Open Science subject to minor revision in accordance with the referee suggestions. Please find the referees' comments at the end of this email.

The reviewers and Subject Editor have recommended publication, but also suggest some minor revisions to your manuscript. Therefore, I invite you to respond to the comments and revise your manuscript.

- Ethics statement

- Data accessibility

<http://datadryad.org/submit?journalID=RSOS&manu=RSOS-192080.R1>

- **Competing interests**

- **Authors' contributions**

- **Acknowledgements**

- **Funding statement**

Because the schedule for publication is very tight, it is a condition of publication that you submit the revised version of your manuscript before 29-Jul-2020. Please note that the revision deadline will expire at 00.00am on this date. If you do not think you will be able to meet this date please let me know immediately.

- 1) A text file of the manuscript (tex, txt, rtf, docx or doc), references, tables (including captions) and figure captions. Do not upload a PDF as your "Main Document".

- 2) A separate electronic file of each figure (EPS or print-quality PDF preferred (either format should be produced directly from original creation package), or original software format)
- 3) Included a 100 word media summary of your paper when requested at submission. Please ensure you have entered correct contact details (email, institution and telephone) in your user account
- 4) Included the raw data to support the claims made in your paper. You can either include your data as electronic supplementary material or upload to a repository and include the relevant doi within your manuscript
- 5) All supplementary materials accompanying an accepted article will be treated as in their final form. Note that the Royal Society will neither edit nor typeset supplementary material and it will be hosted as provided. Please ensure that the supplementary material includes the paper details where possible (authors, article title, journal name).

on behalf of Professor Mirella Lapata (Associate Editor) and Marta Kwiatkowska (Subject Editor)
openscience@royalsociety.org

Associate Editor Comments to Author (Professor Mirella Lapata):

Comments to the Author:

Dear Authors,

I am very pleased to accept your paper for publication. Both reviewers agreed that you have improved the clarity of your paper and took on board most of their suggestions. They make some minor recommendations which I believe could benefit the presentation of the manuscript, and would not be very time consuming. I would recommend that you take these into account when preparing the final version of your manuscript.

Congratulations!
Mirella Lapata

Reviewer comments to Author:
Reviewer: 1

Comments to the Author(s)

The authors significantly improved the clarity of the manuscript with respect to the methods. Their response letter also greatly helped my understanding of the inherent limitations of the data,

and especially the necessity of performing aggregate, rather individual-level analyses. I see the limitations adequately reflected in the manuscript, and I believe that the main contributions of the work stand even in the face of such limitations, given the generality of the proposed framework.

Given the access to the data and the extended clarification of the measures, a minor point became less clear to me, specifically in how errors are computed in figure 3 and sec 4d. To my understanding, the absolute error to validate the model is computed on the raw values for each measure. If this is the case, I am not sure that errors lay on a $[0, 1]$ scale, as reported in the abstract, even for ratio features like Agg./Friend. If I understood correctly, and if no normalization of the raw values is performed, the wider error bars for fig 3b-c may be due to unequal scales or variances in the measures, rather than measurement errors of the tool.

I would also benefit from a slightly more detailed description of table 4. Are performance metrics computed in terms of exact match with the ground truth, e.g. if the number of all characters inferred matches with that of the ground truth, regardless of the magnitude of the discrepancy?

In all other respects, I find the paper improved, therefore I confirm my previous positive judgment.

Reviewer: 3

Comments to the Author(s)

In the paper titled "Our Dreams, Our Selves: Automatic Analysis of Dream Reports", the authors have designed a NLP tool to score dream reports in Hall & de Castle's scale. The tool identifies proper nouns for characters, and verbs to for interactions between them. I find the paper very interesting, and the accompanying website <http://social-dynamics.net/dreams/> visually beautiful. I also checked the authors' responses to earlier reviews and could see the additions made to the paper. Overall, I think the paper is ready for publication. I've a few minor comments which I would like the authors to consider while preparing the camera ready version.

1. They should discuss a little about the continuity hypothesis in the introduction. It presently appears without the underlying context.
2. I thoroughly enjoyed the background section, but felt that it can be better organized with subsection titles.
3. After presenting necessary background, can all five hypotheses come together?
4. Merge sections 6 and 7 into a unified "Concluding Discussion" section.
5. The xtick labels in Figure 3 are difficult to see. Same is true for the greyed words and legend in Figure 2. Repeating "The bars corresponding to the values that do not differ significantly from the norm ($p > 0.01$) are grayed-out" in every caption doesn't look good.
6. I would request the authors to give a careful pass end-to-end. The grammatical errors need to be fixed. For example: "and they not captured other aspects" (page 3, line 31).

Finally, I enjoyed reading the paper and think that it makes important contribution. Hence, I recommend acceptance of the paper.

Author's Response to Decision Letter for (RSOS-192080.R1)

See Appendix B.

Decision letter (RSOS-192080.R2)

Dear Dr Aiello,

It is a pleasure to accept your manuscript entitled "Our Dreams, Our Selves: Automatic Analysis of Dream Reports" in its current form for publication in Royal Society Open Science.

on behalf of Professor Mirella Lapata (Associate Editor) and Marta Kwiatkowska (Subject Editor)
openscience@royalsociety.org

Appendix A

Response to the reviews of paper: “Our Dreams, Our Selves - Automatic Analysis of Dream Reports”

Alessandro Fogli, Luca Maria Aiello, and Daniele Quercia

We would like to express our sincere thanks to the Associate Editor and the reviewers for their very detailed and constructive comments. We have worked to address all their concerns in the revised version of the manuscript. Below, we explain how we have done so.

Summary of Associate Editor’s requests

The revised version which should address the following:

- 1. Give a more detailed description on the coding system*
- 2. Clarify issues with provenance of the dataset, providing statistics for age, sex, etc.*
- 3. Evaluate subcomponents of proposed method and in general fix issues with evaluation raised by both reviewers (e.g., remove confounds in analysis).*

We have thoroughly revised the paper and followed the guidance provided. A summary response to those three points:

1. We have extended our description to all the components of the coding system and explained why we focused on a subset of its dimensions.
2. We expanded the statistics on the dataset by adding, as requested: the number of individual dreamers over time, a breakdown by gender, and the relationship between different dream sets and the sets of dreams from Dreambank.
3. As recommended by the reviewers, we: *i)* provided an evaluation of precision and recall of the individual components of the text processing pipeline; and *ii)* aggregated the results by gender and by decade to disentangle the observed trends by possible confounders.

A detailed breakdown of the actions taken can be found next.

Requests from Reviewer 1

Computing h scores with respect to dataset-specific averages (Section 4.c.vi) comes with a few assumptions and limitations. There is limited information on to what extent the dataset is representative of the US population, comprised of emotionally stable adults, or psychologically healthy individuals, which are assumptions guiding the interpretation of the h values in section 5. Furthermore, although this process allows for relative comparisons within it, it somewhat hinders robustness with respect to new data from other domains. I am not fully convinced that switching the reference normative set is an advantage for the tool in general, beyond proving the research hypotheses of the paper.

Unfortunately, no dream report dataset publicly available to date allows for the computation of norms that would be representative of the US population. The normative set coded by Hall and Van De Castle included only dreams from young college students who volunteered at the time of data collection. Given that, we have computed alternative norms based on a larger and hopefully more representative sample. However, we cannot claim that such larger sample reflects the country's distributions of socio-demographic features—on Dreambank there is simply not enough information about the dreamers to even attempt making such an assessment. We reported this limitation more clearly in the new Discussion section. Ultimately, the normative values are just used to rescale the h -values computed by our tool; if and when better normative values will be found by dream researchers, those could easily replace the ones we used in our pipeline.

Second, the authors do not give information on the skew of individual dreamers' proficiency in the dataset, and do not account for individual differences in the analyses. Controlling for individual differences would better fit the analyses, since ultimately the continuity hypothesis applies to individuals, and if dreamers like Izzy dominate the datasets it would be hard to interpret the findings.

We agree with the reviewer that assessments at individual level are the most valuable, and that's why we focused on Izzy and the war veteran as two interesting case studies. Unfortunately, the individual assessment of dreamers cannot be extended to the whole Dreambank dataset because many dreams are associated with groups of people (e.g., young male students, brides, Peruvian women) rather than being linked to individual identities. This limitation does not make it possible to systematically calculate the h -values as macro-averages of averages computed at individual level. We made sure to highlight this limitation in the Discussion section. However, following the suggestion of Reviewer 2, we attempted to mitigate this issue by complementing the h -values computed for testing the Hypotheses 1 and 5 (those that use the whole population of healthy dreamers) with h -values that are macro-averages of the averages obtained for each decade and for each of the two reported genders. By weighting each decade equally, for example, the contribution of Izzy's dreams on the final values are heavily discounted compared to our previous calculation. The new values we obtained, albeit different from the old ones, exhibit trends that are consistent with the results previously presented. We added the new results in Figures 4 and 7.

Requests from Reviewer 2

It would be great if the authors could give a brief description of the coding system and also explain why they chose to only code the dream reports along their selected three categories. Could the exclusion of the other categories hinder the inferences that they can make. I believe such a description would make the method section more complete and make it easier for non-experts to also follow their paper.

We added a complete description of all Hall-Van de Castle coding categories in Section 4(a). We report them here for the reviewer's convenience:

- *Characters*: people, animals, and other figures featured in the dream;
- *Interactions*: social interactions among characters (e.g., kissing);
- *Emotions*: emotions experienced by the characters or denoting the situation (e.g., sadness);
- *Activities*: physical actions that characters perform and sensory experiences (e.g., smelling);
- *Striving*: Success and failure of characters in carrying out their activities;
- *(Mis)fortunes*: Fortune and misfortunes that happen to the characters, possibly as result of their actions;
- *Settings and objects*: Physical surroundings or objects present in the scene (e.g., outdoors, a weapon);
- *Descriptive Elements*: Attributes and qualities of objects, people, and actions (e.g., color, size, speed);
- *Food and Eating*: Presence of food or act of eating;
- *Elements From the Past*: Characters or elements belonging to the dreamer's past (e.g., younger self).

In practice, these categories are not of equal importance in capturing the psycho-pathological value of the dream's content. Dream scientists determined that the three categories of Characters, Social Interactions, and Emotions are the most valuable ones (see "*The scientific study of dreams: Neural networks, cognitive development, and content analysis*" by William Domhoff, 2003) and are usually more informative than all the remaining ones combined. Those three categories are sufficient to verify our hypotheses, therefore we chose to focus on them only.

On page 7, lines 48-51, authors indicate that they also performed the paired t-tests. I do not see where they have reported these results in the manuscript.

All the h -values we reported in Figures 4, 5, 6, and 7 are complemented with t -tests. The p -values we obtained are all < 0.01 except for those indicated by the grayed-out bars. We made sure to state this clearly in the text as well as in the figures' captions.

It is not immediately clear to me which dreams on dreambank.net belong to which of author's 7 datasets. It would greatly help reproducibility if the authors would provide this information. Perhaps the authors have included it at <https://doi.org/10.5061/dryad.n02v6wwsq>, however I could not access it (DOI not found error) to verify it.

We apologize for the broken DOI: it appears that Dryad does not activate the DOI before publication, but it provides a temporary “reviewer URL” that allows to download the data: <https://datadryad.org/stash/share/WaaWXMsr3Gv0JYdtRyRgah-rQsKFiGqrvGtAd5ieaY8>, which should be accessible. In the Appendix, we added Table 4 that lists all the dream groups from Dreambank corresponding to each of the dream sets we studied.

I would have also liked to see an evaluation at each step of the automation pipeline to better understand which of the automation steps work well and which don't. This is especially important since different dictionary approaches and NLP methods have been employed to infer the different measured quantities and its not clear that they all work equally well. For instance, since the authors identify the different characters (C_{people} , $C_{animals}$, and $C_{fictional}$), and the normative set contains these manual annotations for the characters in about 1000 dream reports. Authors could present the precision and recall numbers for the different types of characters. Similar method could be used to measure and report the precision and recall using the normative set for other measured quantities like C_{males} , $C_{females}$, C_{dead} , $I_{friendly}$, $I_{aggression}$, I_{sexual} , $W_{negative}$ and $W_{positive}$. A table summarizing these results would help future researchers to estimate how well the tool would work for other related scenarios for dream analysis.

As the reviewer suggested, we added a table (Table 4 in the new manuscript) that reports the precision, recall, and F1 score for the different characters, interactions, and emotions categories.

It would be great if the authors could also report the size of the final lists they used for identifying characters, character's properties, social interactions and emotions.

In Section 4.c we added the size of the word lists we have used.

Why does the size of the normative set differ when reported in different parts of the paper – 1000 on page 9, 600 on page 13.

The normative set contains 981 dreams (roughly 1000). We corrected the second figure, which was wrong.

I would encourage the authors to take a bit more care in explaining their evaluation. While Figure 3 is quite informative, I think it is not sufficiently explained in the text (especially Figure 3A). A bit more explanation would help the readers to follow along more easily. I also wonder what the errors look like for the dataset. It would be great if they authors could include the normative set results (which they call literature) in Figure 3B and 3C to help others contextualize their evaluation results. Also, please use consistent terminology and dataset names.

We thank the reviewer for spotting the inconsistency in the evaluation description about using different dream sets for measuring our tool's error. Figure 3 shows the results obtained from *all* the available annotated dreams. We correctly stated it at the beginning of Section 4(d), but later on in the same section we reported a contradictory statement, which obviously created confusion. We revised that subsection to clarify the procedure.

The results for 2 of their 5 hypotheses regarding adolescence (1 dreamer) and post war trauma (1 dreamer) are based on the dreams of a single dreamer and as the authors themselves point out, this makes it hard to verify these hypotheses with confidence. For another hypothesis regarding blind people, again the results are based on 16 dreamers (though a higher number of dreams), and I am not sure if this number is sufficient to consider the hypothesis verified.

The low number of dreamers in these three categories is given by the restricted scope of the dataset—a limitation that unfortunately we cannot overcome in this study. Data augmentation in this domain is particularly challenging: it is very expensive to collect dream reports from the same person for multiple years or to gather reports from individuals with rare conditions such as war-induced PTSD. Part of the motivation to build our automatic dream analysis tool comes from the need to enable technologies that can incentivize more people to report their dreams on a regular basis. In the revised version, we made sure to emphasize this limitation in the Discussion section by stating that our results offer supportive rather than conclusive evidence for the continuity hypothesis.

Page 15, line 39, authors mention that “based on available descriptions about the dreamers, the distribution of ages in the two sets should be comparable...”. I would be more convinced if the authors could include what information available in the descriptions convinced them of the similar age distributions.

Demographic information about Dreambank dreamers is patchy and incomplete. The identity of many dreamers is unknown, as several individuals are sorted in groups (e.g., a group of Peruvian women). When the identity is known, demographic information such as their age is not always provided. Our assessment of the comparability of the age distribution comes from a ballpark estimation of the number of children, adults, and elderly men and women that we could infer from the available descriptions on Dreambank. However, the data does not allow us to back this statement with a grounded statistical assessment. We therefore removed that statement and reported this limitation in the Discussion section.

I noticed that the male and female datasets include dreamers from a wide range of decades 1910-2010. Are the reports from males and females equally distributed across the decades, otherwise the difference in the decades might also lead to the differences in the observed level of aggression etc.

It would be helpful to also report the number of dreamers for each decade. It is unclear to me, how many people's dreams are utilized to test hypothesis 5. Also is the gender distribution similar for the different decades? Otherwise, the increased amount of aggression for 1960's might just be a consequence of mostly men's dreams having being analyzed.

In general, I am not sure if the effects of confounding variables (for instance age or decade while examining hypothesis 1 or gender distribution for different decades while examining hypothesis 5) have been sufficiently disentangled or controlled for.

We added more detail to the distribution of the number of male and female dreams (and dreamers) in Figure 1. Both the overall volume of data and the gender ratio vary across decades.

To check whether this imbalance impacted the results when testing Hypothesis 1 (differences between male and female dreams), we re-calculated the h -profiles using a different approach. Previously, we computed h -values for each dream and then we averaged the resulting values over all dreams. In the revised version, we complemented those average value with a new measurement: we first group dreams by decade (starting from the 1930s, as earlier decades are too sparse), compute the per-decade averages, and then macro-average across decades. By doing so, we weight every decade equally, regardless of their dream count, thus mitigating the imbalance issue. We overlay the newly-calculated macro-averages as bullets on top of the Figure 4 histogram, for the sake of comparison. The trends are quite similar to the ones we got with the micro-averaging approach.

In a similar way, to check if our findings related to hypothesis 5 were affected by the uneven distribution of male and female across decades, we re-computed the h -values of different decades by considering only male (M) or only female (F) dreamers. The new results are shown on top of the histogram in Figure 7 with small M and F markers. We observed that the aggression for both men and women steadily decreased over the decades, even though the male values are always higher than those measured of female's dreams.

For the hypothesis 5, how did the other aggressive interaction features vary with the decades.

We complemented Figure 7 with a new plot showing the trend of *Aggression %* over time. The absolute h -values are lower than what we found for the *A/C Index* and the confidence intervals are wider, yet the trend is consistent.

Some of the positive comments...

(AC) The manuscript is very interesting and we would like it to appear in the journal

(R1) I thank the authors for the interesting read. I overall found the paper engaging and the results fascinating.

(R2) Utilizing of NLP techniques to assist in manually intensive tasks in non-computational fields of study could help to scale up the size of studies typically performed and the authors present such an application of NLP to analyze dream reports. I like the simplified research hypotheses that the authors have formulated to find evidence for 'continuity hypothesis'.

We want to express our sincere thanks to the Associate Editor and to the Reviewers for the positive comments above, and hope they will find the new version of the paper improved.

Appendix B

Response to the reviews of paper: “Our Dreams, Our Selves - Automatic Analysis of Dream Reports”

Alessandro Fogli, Luca Maria Aiello, and Daniele Quercia

We would like to express our sincere thanks to the Associate Editor and the reviewers for deeming the paper suitable for publication. Below, we summarized the changes we made to the manuscript to address their latest comments.

Requests from Reviewer 1

To my understanding, the absolute error to validate the model is computed on the raw values for each measure. If this is the case, I am not sure that errors lay on a $[0, 1]$ scale, as reported in the abstract, even for ratio features like Agg./Friend. If I understood correctly, and if no normalization of the raw values is performed, the wider error bars for fig 3b-c may be due to unequal scales or variances in the measures, rather than measurement errors of the tool.

Yes, the reviewer is right about this point. Technically, some of the measures are not bounded between 0 and 1. In practice, both the ground truth and the scores computed by our tool are almost always in $[0,1]$. The measure that deviates most from this range is the A/C Index: it is greater than 1 in 6% of the cases in the ground truth and in 3% of the cases according to our tool. We removed the claim that the errors are bound in $[0,1]$ and, in the Evaluation section, we added a short discussion on the ranges.

I would also benefit from a slightly more detailed description of table 4. Are performance metrics computed in terms of exact match with the ground truth, e.g. if the number of all characters inferred matches with that of the ground truth, regardless of the magnitude of the discrepancy?

Indeed, the table needed a clearer explanation which we provided in the revised version. The precision and recall values in the table refer to the set of elements that are correctly identified in the dream. For example, if the dream includes 4 male characters and our tool identifies two of them correctly, the recall will be 0.5. We denote with C_{Male} the *set* of male characters and with $|C_{Male}|$ the *number* of male characters. This part of the evaluation is concerned with checking whether the tool can identify the elements in those sets, rather than producing the same count.

Requests from Reviewer 3

They should discuss a little about the continuity hypothesis in the introduction. It presently appears without the underlying context.

We expanded the description of the continuity hypothesis in the introduction.

I thoroughly enjoyed the background section, but felt that it can be better organized with subsection titles.

We split the background section into smaller subsections.

After presenting necessary background, can all five hypotheses come together?

We now provide the summary of the five hypotheses at the end of Section 3.

Merge sections 6 and 7 into a unified "Concluding Discussion" section.

We merged the last two sections into one, as suggested.

The xtick labels in Figure 3 are difficult to see. Same is true for the greyed words and legend in Figure 2. Repeating "The bars corresponding to the values that do not differ significantly from the norm ($p < 0.01$) are grayed-out" in every caption doesn't look good.

We enlarged the fonts in Figure 3 and make the gray text darker in Figure 2. We kept the explanation in the captions to keep the figures as much self-contained as possible.

I would request the authors to give a careful pass end-to-end. The grammatical errors need to be fixed. For example: "and they not captured other aspects" (page 3, line 31).

We revised the paper thoroughly to fix typos and grammatical errors